# Is the Next Winter Coming for AI?
# The Elements of Making Secure and Robust AI

## Abstract

While the recent boom in Artificial Intelligence (AI) has given rise to the technology's use and popularity across many domains, the same boom has exposed vulnerabilities of the technology to many threats that could cause the next "AI winter". AI is no stranger to "winters", or drops in funding and interest in the technology and its applications. Many in the field consider the early 1970's as the first AI winter with another proceeding in the late 1990's and early 2000's. There is some consensus that another AI winter is all but inevitable in some shape or form, however, current thoughts on the next winter do not consider secure and robust AI and the implications of the success or failure of these areas. The emergence of AI as an operational technology introduces potential vulnerabilities to AI's longevity. The National Security Commission on AI (NSCAI) report outlines recommendations for building secure and robust AI, particularly in government and Department of Defense (DoD) applications. However, are they enough to help us fully secure AI systems and prevent the next "AI winter"? An approaching "AI Winter" would have a tremendous impact in DoD systems as well as those of our adversaries. Understanding and analyzing the potential of this event would better prepare us for such an outcome as well as help us understand the tools needed to counter and prevent this "winter" by securing and robustifying our AI systems. In this paper, we introduce the following four pillars of AI assurance, that if implemented, will help us to avoid the next AI winter: security, fairness, trust, and resilience.

## 1 Introduction

In "A Choice of Catastrophes" [1], Isaac Asimov outlines an extensive array of possibilities that could result in the "end of the world". Some are inevitable, but some are avoidable with the right knowledge, precautions, and action. Using this lens, are there lessons we can learn that extend to the "end of AI"? What are the most likely ways that artificial intelligence (AI) will succumb to its next "winter"? What can we learn from previous AI Winters to shed light on current progress and possible pitfalls that lie before us? Recent work in adversarial machine learning has shown us that AI can be very vulnerable to seemingly benign changes in inference data, for example. What predictions can be made with regard to AI security with these recent papers demonstrating successful attacks on AI, but also successful defenses against those attacks? With the push for fielding AI systems gathering steam in industry and the DoD, these questions warrant urgent examination.

The field of AI is rapidly growing, and deployment of AI-enabled systems is gaining traction at nearly the same pace. These deployments also include operations and applications within the U.S. government and DoD, so trust in the security and robustness of these systems is paramount. Similarly, AI is being deployed in health, transportation, automation, and essentially every technology

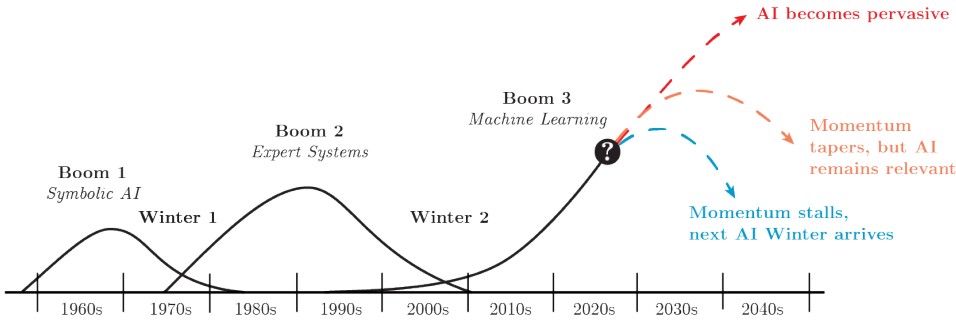

Figure 1: AI Winters

and infrastructure imaginable. However, AI was not developed overnight, nor has it been successful at every turn in its history. Conversely, there have been several "AI Winters" over the past decade where an explosion of interest, funding, and progress were stopped in their tracks. The causes of these "winters" are many but studying their causes and effects could help us with any looming "winter" in our current AI future. Also, new threats are emerging in the area of adversarial machine learning that could have the potential to halt AI progress if they are not properly studied and averted. Since the last AI Winter, we also have new approaches and tools to help us on the journey to secure and robust AI, such as lessons learned from cybersecurity and from red-teaming systems and applications.

## 2 Related Work & The Four Seasons of AI

While many authors have addressed the concept of the AI winter [2–4], the work of Haenlin and Kaplan [5] introduces a more complete picture with a summary of the "four seasons of AI". In the AI Spring, the authors pinpoint the roots of AI to the Isaac Asimov article *Runaround* where Asimov introduces the infamous *Three Laws of Robotics*. His work inspired generations of scientists in computer science, AI, and robotics. Marvin Minsky, who later founded the MIT AI laboratory, was among those scientists inspired by Asimov. Around the same time, Alan Turing would publish his article "Computing Machinery and Intelligence" which established the benchmark Turing Test for identifying and evaluating intelligence in an AI system. Credit for the words Artificial Intelligence is given to Marvin Minsky and John McCarthy who hosted the first workshop on the topic at Dartmouth College in 1956. Following the workshop, AI experienced its first summer, resulting in two decades of success both in funding and in technological progress. One example was the ELIZA computer program which was one of the first natural language processing tools that attempted to pass the Turing Test. Famously, partly due to the successes of AI in this first summer, Minsky predicted in 1970 that "a machine with the general intelligence of an average human being could be developed within three to eight years." Obviously this was not the case and just three years later, AI would experience its first winter. British mathematician James Lighthill published a report that questioned the optimistic outlook by Minsky and others and stated that AI would only achieve "experienced amateur" status in games and would never achieve common-sense reasoning. Subsequently, the British government drastically reduced support for AI research and the U.S. government would follow suit. The past two decades of the "AI Fall" have seen the "harvest of the fruits of past statistical advances" beyond Expert Systems that were developed in previous AI summers. Visually, the seasons of AI can be seen in Grudin's work where he connected the history of AI and human-computer interaction (HCI) [6]. Present day advances in artificial neural networks have driven the vast majority of successes in AI. However, the future of AI is the main interest of this article. Haenlin and Kaplan outline a need for regulation in their article in the following themes; data bias, black box systems, workforce changes, and privacy. Data bias can cause unintended and harmful outcomes when used to develop AI systems. However, developing commonly accepted requirements for training data and methodologies may be more effective than regulating the AI itself. The concern with black box systems is that, in the context of consequential use, we need to understand how decisions and recommendations are made from these systems. To avoid disruption in the workforce that will undoubtedly be affected by the advances of AI technologies, retraining of the workforce towards new jobs that cannot be automated is one direction to consider. Finally, there will certainly be a need to

balance personal privacy concerns with the economic growth and technology gains we will see as AI continues to gain success. However, a much broader question is posed by the authors for future AI systems of "how do we regulate a technology that is constantly evolving".

Prior to the recent resurgence of AI, several researchers reflected on funding and interest that was in flux in the early 2000s. In 2005, Waltz noted the changing landscape of Association for the Advancement of Artificial Intelligence (AAAI) over the years and in particular how dwindling attendance was in a large part due to newer conferences that were spun off, such as knowledge discovery and data mining (KDD), and other conferences in natural language processing, vision, robotics, and learning [7]. He also notes that, even in 2005, the most recent AI winter was a "distant memory" which had been eclipsed by the tech bubble of the early 2000s. He also predicted at that time that AI was "entering a new golden age".

In 2006, John McCarthy published a short, but insightful manifesto on the future of AI [8]. In that article, McCarthy points to "logical AI" as the "best hope for human-level AI", but also states that approaches "such as neural nets may also work". He also points out that the "AI winter was dominated by people who lost money in companies" and warns that "AI research should not be dominated by near-term applications". These are certainly wise recommendations as we navigate the current AI landscape of research and industry investment. Also in 2006, Grosz stressed the importance of diversity in the field of AI when it comes to modeling intelligence and the "need for people who focused on building systems to respect theories and for those developing theories to appreciate the challenges of building systems, and for us to collaborate with one another both in research and in supporting our field" [9]. Further she argues for the collaboration of those throughout different areas of computer science so that AI capabilities would be "designed as parts of systems."

In 2007, James Hendler asks "Where are all the Intelligent Agents [10]?" After more than a decade of work, such as that published at the International Joint Conference on Autonomous Agents and Multiagent Systems (AAMAS), Hendler claims to "see no evidence for the imminent widespread use of" agents in applications like web development. This speaks to the slow adoption of AI in industry before the most recent "AI summer". In 2008, James Hendler follows up his 2007 article with thoughts on how to avoid another AI winter [11]. Having lived through the AI winter in the 80's, he warns that we might be seeing early signs of "a change in the weather". He astutely points to the growing trend at the time that "funding for university researchers has all too often come with an expectation of fast transitions to industry". On "weatherproofing" against a possible AI winter, Hendler suggests that we embrace operational and applied AI and "ensure we acknowledge the success we see."

In more recent times, several researchers have shared their viewpoints on past and future AI winters and their attributes. Duan, Edwards, and Dwivedi [12] raise the ethical and legal issues stating that "rapid advances in AI are raising serious ethical concerns." The authors point out the role that the government plays in addressing ethical and legal concerns on the use of AI and that "it is imperative that more research must be carried out on the role of the government in shaping the future of AI." They make the following proposition for consideration on this topic: "government plays a critical role in safeguarding the impact of AI on society."

In Floridi's article [13]: "The risk of every AI summer is that over-inflated expectations turn into a mass distraction". There are three possibilities with AI solutions as compared to current or previous solutions. They can *replace* "as the automobile has done with the carriage"; *diversify* "as did the motorcycle with the bicycle", or *complement* or *expand* them, "as the digital smart watch has done with the analog one." A key question to ask going forward: "are the necessary skills, datasets, infrastructure, and business models in place to make an AI application successful?" With a more cautionary view, Hofstetter, Koumpis, and Chatzidimitriou argue in their 2020 artcle [14] that "most companies and industries are not ready for ML" and that ML is often "seen as a magic bullet that can solve anything, which is simply not true." They also argue that companies are throwing ML at problems that are extremely difficult, "like predicting the stock market." The authors stress that companies and practitioners of AI and ML need to ask the right questions, such as "Why Data Science? Why AI? Why ML?", when approaching a problem and potential use of the technology.

In addition to the above challenges, there is further evidence of the difficulty in the implementation and establishment of the right government bodies and authorities to oversee the development of AI. For example, after only four years of existence within the DoD, the Joint AI Center (JAIC) will cease to exist and instead be rolled into the newly created Chief Digital and Artificial Intelligence

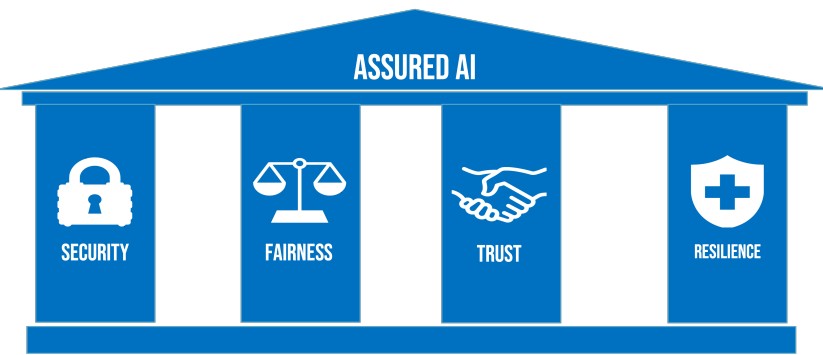

Figure 2: The Four Pillars of AI Assurance

Officer (CDAO) [15]. However, the topics of assured AI, which includes security, fairness, trust, and resilience, are top of mind for the DoD and many government bodies as the pressure and need to accelerate the adoption of AI continues to mount.

## 3  AI Assurance Framework

To address the challenges of making AI truly operational, particularly for consequential uses of AI, we propose a framework for AI assurance as shown in Figure 2. Supporting evidence of the need of such a framework comes from many sources. First, the National Security Commission on AI (NSCAI) Final Report [16] lists several recommendations "to accelerate AI innovation to benefit the United States and to defend against the malign uses of AI." With regards to AI assurance as a whole, the NSCAI states that "there has not yet been a uniform effort to integrate AI assurance across the entire U.S. national security enterprise." Further, the NSCAI Final Report enumerates several recommendations, including the following. When discussing the potential security risks of operational AI, the NSCAI recommends that the Department of Defense (DOD) and related government bodies "consider establishing government-wide communities of AI red-teaming capabilities that could be applied to multiple AI developments."

Similarly, the DoD recently released the "U.S. Department of Defense Responsible Artificial Intelligence Strategy and Implementation Pathway" [17] which outlines the DoD's AI Ethical Principles of 'Responsible', 'Equitable', 'Traceable', 'Reliable', and 'Governable' AI. As part of the DoD's Chief Digital and Artificial Intelligence Office (CDAO) Responsible AI (RAI) strategy and implementation, they list the following as components: RAI Governance, Warfighter Trust, AI Product and Acquisition Lifecycle, Requirements Validation, Responsible AI Ecosystem, AI Workforce.

In response to the challenges of AI for Europe, the European Commission created the European High-Level Expert Group on AI (AI-HLEG) [18] which defines three main components of trustworthy AI, which should be met throughout the system's entire life cycle:

1. lawful, complying with all applicable laws and regulations;

2. ethical, ensuring adherence to ethical principles and values;

3. robust and secure, both from a technical and social perspective since, even with good intentions, AI systems can cause unintentional harm

In a survey of AI enabling technologies, Gadepally et al. [19] list robust and trusted AI as foundational technology underpinnings of AI development. The authors identify explainabliity, measures of effectiveness, verification and validation, and the ethical use of AI as components to robust and trusted AI. In 2015, which some would consider the early days of the current boom in AI, Russell, Dewey, and Tegmark penned "Research priorities for robust and beneficial artificial intelligence" [20] which outlined both short-term and long-term priorities at the time. Very similar themes of ethics research, research for robust AI, verification, and security appear in the article as research directions to avoid "potential pitfalls."

Additionally, several companies, such as Google, Microsoft, Facebook, and IBM have outlined their own versions of responsible, ethical, and trustworthy AI strategies [21].

The AI assurance framework outlined in this section consist of four pillars to address the challenges we all face in operationalizing consequential uses of AI: Security, Fairness, Trust, and Resilience. Each of the four pillars is outlined in more detail below. In order to fully implement such a framework, however, it will require BOTH a complement of technical solutions as well as effective governance.

## 3.1 Security

Adversaries are developing and acquiring ever more sophisticated AI-driven platforms, dramatically increasing their ability to rapidly carry out their mission. The U.S. government and their partners are increasingly relying on intelligence derived from AI models and partnering with non-traditional actors to deploy these capabilities. AI algorithms have a unique attack surface that represents both an opportunity to disrupt our adversaries' events chains and a risk in the increased attack surface on our systems. Understanding and mitigating risks in AI security is paramount to the proliferation of AI in real-world, consequential applications of the technology.

Adversarial attacks on machine learning, where, for example, an input is perturbed at inference time to induce an erroneous decision, pose real threats to deployed models. The number of academic papers on this topic on both the attack and defense perspective has exploded in recent years and there are several surveys that give an overview of the research [22–24]. From white-box attacks, where the adversary has complete access and knowledge of the system, to black-box attacks, which assume no adversary knowledge of the system, adversarial threat models pose various levels of threat to real-world AI systems. The NSCAI Final Report recommends that we "focus more federal R&D investments on advancing AI security and robustness". One effort to address this security gap is the Adversarial Threat Landscape for Artificial-Intelligence Systems (ATLAS) [25], which is a "knowledge base of adversary tactics, techniques, and case studies for machine learning (ML) systems based on real-world observations, demonstrations from ML red teams and security groups, and the state of the possible from academic research." ATLAS is made possible by a consortium of partners, such as MITRE, IBM, Microsoft, and NVIDIA. By sharing the tactics, techniques, and procedures used by adversaries to attack real-world systems, along with case studies depicting attacks in detail, the community can learn system vulnerabilities as well as defense mechanisms to such attacks.

To further support research needed on AI security, a recent article focusing on ML safety [26] points out four unsolved problems that need to be addressed by researchers and practitioners: withstanding hazards ("Robustness"), identifying hazards ("Monitoring"), reducing inherent model hazards ("Alignment"), and reducing systemic hazards ("Systemic Safety"). Additionally, researchers and practitioners in the cybersecurity domain have been paving a path to a more holistic approach to security by viewing them through a lens of build, attack, and defend teams [27, 28].

## 3.2 Fairness

While many definitions of "fairness" exist, especially as related to AI, the umbrella we are viewing the term is broad and inclusive. We follow the broader concept of fairness to include ethics, accountability, transparency, bias, equity, and justice, as Birhane et al. [29] describes. John-Mathews, Cardon, and Balagué [30] also have a similar umbrella for their definition including fairness, privacy, and transparency as a basis for ethical development of AI [30]. Nelson [31] argues for primary tenets to evaluate bias in ML models: transparency, trust, fairness, and privacy.

A recent survey on bias and fairness in ML [32] explores real-world cases of "unfair" uses of ML algorithms. The authors also describe the different types and sources of biases that can occur and how fairness has been operationalized. The authors of "Auditing the AI auditors: A framework for evaluating fairness and bias in high stakes AI predictive models" [33] take a slightly different approach from a point of view of measuring fairness and bias using research from the measurement of psychological traits. In defining fairness and bias in their work, they look first to "individual attitudes" and a "framework consisting of distributive, procedural, and interactional justice perceptions." Beyond the individual, the authors also work to define fairness and bias "through the lens of legality, ethicality, and morality." The third lens they use for these definitions is based on embedding these meanings in technical domains, or essentially basing the definitions in statistics.

Silberg and Manyika [34] give their own definitions of fairness and bias and also lay out a framework for maximizing fairness and minimizing bias in AI. The framework consists of: awareness of bias in AI, particularly in contexts in which there is a high risk of bias; establish best practices to test for and mitigate bias; engage in "fact-based conversations about potential biases in human decisions"; invest in bias in AI research and adopt a multidisciplinary approach; and invest more into the AI field and diversification of the field itself.

Bellamy et al. introduces IBM's toolkit for detecting and mitigating algorithmic bias, called AI Fairness 360 [35]. This popular toolkit has been cited many times and used in several real-world applications of measuring bias, such as the companion book [36], which focuses on how teams can mitigate unfair machine bias by using the open source tools available in AI Fairness 360. Additionally, there is a freely available course called "Introduction to AI Fairness" [37] that covers recent developments in algorithmic fairness, including definitions of fairness like those we have discussed above, their corresponding quantitative measurements, and ways to mitigate biases.

AI Fairness 360 is a great example of real-world tools that will help us explore and mitigate bias and fairness issues with AI to better understand this pillar of AI assurance.

## 3.3 Trust

When discussing the development of trustworthy AI, explainability as well as predictability are often used in its definition. Hamon, Junklewitz, and Sanchez outline in their report on "Robustness and explainability of artificial intelligence" [38] three important topics on the topic of trust: transparency of models, reliability of models, and protection of data in models. Jha presents a tutorial [39] on their Trusted, Resilient and Interpretable AI framework called Trinity being developed at SRI to tackle real-world problems and challenges related to trust in AI.

In response to the National AI Research and Development Strategic Plan [40], the National Science Foundation (NSF) created several new AI institutes, one around the theme of Trustworthy AI, that is expected to fund several universities and projects later this year. These types of research funding opportunities will be paramount for the future work in trust and explainability for AI.

## 3.4 Resilience

The final pillar of our AI assurance framework is resilience, which is usually accompanied by the concept of robustness in most definitions. The themes of test and evaluation (T&E) and validation and verification (V&V) are usually associated with resilient and robust AI as well. While a lot of research and resources have gone into the testing and verification of autonomous systems that use applications of AI [41], the application of T&E and V&V methodologies to modern AI systems are less studied.

As a reminder, both the works by Gadepally et al. [19] and Russell, Dewey, and Tegmark [20] called for measures of effectiveness, verification and validation, and research in robust AI to address the resilience gap in AI applications. In the work of Brown, Curtis, and Goodwin [42], the authors outline their "Principles for Evaluation of AI/ML Model Performance and Robustness". They state that in order for an AI/ML model to be considered robust, it should exhibit properties of generalization, or "good performance on data that is drawn from the same distribution as the training data but not used explicitly during training", and robustness, or the model's ability to "maintain performance, with graceful degradation, as the unseen test data becomes increasingly different from the training data."

In [43] Jin et al. summarize a workshop held on the resilience of cyber-physical systems (CPS) which highlighted four promising themes for CPS research: Resilient Topologies of Sensors and Hardware, State-of-the-Art Modeling and the Digital Twin, Machine Learning and Artificial Intelligence, and Energy Networks and the System of Systems.

Resilience and robustness are of crucial importance to the development of AI in real-world systems. Industry and government institutions are focusing more effort in recent years on this important and challenging topic.

## 4 Conclusion

In this paper we have outlined the history of AI winters along with a summary of past causes of these winters. We defined AI assurance as having four pillars of security, fairness, trust, and resilience to tackle the many issues exposed by past AI winters as well as current adoption issues for uses of AI in consequential applications. We have shown that these four pillars encompass many of the issues brought forth, such as ethics, robustness, bias, and explainability. Having a common language and lexicon when discussing these challenges is extremely important. As we as a community continue to build out the strategic elements and the tools and metrics to measure AI assurance, we will pave a path to increasing adoption of AI in real-world applications and help to stave off future AI winters. We believe that by designing and implementing the AI assurance pillars of security, fairness, trust, and resilience, the next AI winter can be mitigated to a reasonable degree.

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
