# OpenReview forum: "Is the Next Winter Coming for AI?The Elements of Making Secure and Robust AI"
_NeurIPS.cc/2022/Workshop/TSRML — TSRML2022_

### Official Review · Reviewer_E54G · 2022-10-17
**Decent survey paper**

**Overall Rating:** 6

**Summary:**

The paper outlines the history of AI winters along with a summary of past causes of these winters. The term winter refers to drop in funding and interest in the technology and its applications. The authors defined AI assurance as having four pillars of security, fairness, trust, and resilience to tackle the issues exposed by past AI winters as well as current adoption issues for uses of AI.

**Strengths:**

The paper is relevant to the work shop, well articulated, easy to understand and gives a lot of background information.
It clearly explains the past reasons and timeline for AI winters while citing the relevant literature.
The related work section is written quite in depth acknowledging the similar work done over the years by past authors and tries to explain the history of AI boom by circling back to 1956 and describing the work of Marvin Minsky.


**Weaknesses:**

While the paper is detailed and thorough it am not sure if the work itself has much significance beyond a review paper. The authors cite relevant literature and talk about how security, fairness, trust and resilience affects AI winters however there isn't a lot novelty to this al together. The authors don't detail the exact XYZ steps the community must take in order to prevent the next AI winter.

**Overall Recommendation:**

Overall, I think it's a decent read as a literature review for understanding the backgrounds of past AI winters and it's enabling factors however as a research paper it lacks a major significance. I would be okay seeing this accepted if literature review/survey paper is within the scope of the workshop.

**Review Confidence:**

3: The reviewer is fairly confident that the evaluation is correct

---

### Official Review · Reviewer_5tFT · 2022-10-19
**Interesting paper but not convincing enough**

**Overall Rating:** 5

**Summary:**

This paper discussed whether the next winter is coming for AI. Specifically, this paper first reviewed the past AI winters. Then, an AI Assurance Framework consisting of security, fairness, trust, and resilience, is proposed to generalize the current issues and crises. At last, the authors claim that by designing and implementing AI assurance, the next AI winter can be mitigated to a reasonable degree.

**Strengths:**

This paper is well-written and well-organized. The authors summarized a lot of high-confidence evidence and comments from third parties (e.g., the government, research institutes, and tech companies). The designed AI assurance framework is quite interesting. We also believe that controllable and robust AI can help deploy AI algorithms into more physical scenarios, which will mitigate AI winter to a certain degree.



**Weaknesses:**

In my personal view, adding more graph data and quantified results would make this article more objective and credible.

**Overall Recommendation:**

Generally, this paper discussed the relationship between Robust AI and AI winter. The authors claim that Robust AI can help avoid AI winter to a certain degree. The proposed framework seems reasonable and can be helpful to the community.

**Review Confidence:**

2: The reviewer is willing to defend the evaluation, but it is quite likely that the reviewer did not understand central parts of the paper

---

### Decision · Program_Chairs · 2022-10-23

Accept